

# Brief communication: Comparison of in-situ ephemeral snow depth measurements over a mixed-use temperate forest landscape

Holly Proulx[1], Jennifer M. Jacobs[1,2], Elizabeth A. Burakowski[2], Eunsang Cho[3,4], Adam G. Hunsaker[1,2], Franklin B. Sullivan[2], Michael Palace[2,5], Cameron Wagner[1]

[1]Department of Civil and Environmental Engineering, University of New Hampshire, Durham, NH, 03824, USA
[2]Earth Systems Research Center, Institute for the Study of Earth, Oceans, and Space, University of New Hampshire, Durham, NH, 03824, USA
[3]Hydrological Sciences Laboratory, NASA Goddard Space Flight Center, Greenbelt, MD, USA
[4]Earth System Science Interdisciplinary Center, University of Maryland, College Park, MD, USA
[5]Department of Earth Sciences, University of New Hampshire, Durham, NH, 03824, USA

*Correspondence to*: Jennifer M. Jacobs (Jennifer.jacobs@unh.edu)

**Abstract.** The accuracy and precision of snow depth measurements depend on the measuring device and the conditions of the site and snowpack in which it is being used. This study compares collocated snow depth measurements from a magnaprobe snow depth probe and a Federal snow tube in an ephemeral snow environment. We conducted three snow depth sampling campaigns from December 2020 to February 2021 that included 39 open field, coniferous, mixed, and deciduous forest sampling sites in Durham, New Hampshire, United States. For all sampling campaigns and land cover types with a total of 936 paired observations, the magnaprobe snow depth measurements were consistently deeper than the snow tube. There was a 12% average difference between the magnaprobe (14.9 cm) and snow tube (13.2 cm) average snow depths with a greater difference in the forest (1.9 cm) than the field (1.3 cm). This study suggests that snow depth measurements using a Federal snow tube can avoid overprobing with an ephemeral snowpack in forested environment.

**Short Summary.** This study compares snow depth measurements from two manual instruments in a field and forest. Snow depths measured using a magnaprobe were typically 1 to 3 cm deeper than those measured using a snow tube. These differences were greater in the forest than the field.

## 1 Introduction

Snow depth is one of the easier snowpack properties to measure in the field and is an observation that can be measured relatively precisely without considerable expertise or expense. Hundreds of snow depth measurements can readily be taken in a single day and automated samplers can substantially increase that number (Sturm and Holmgren 2018). In-situ snow depth observations can be measured manually or automatically. While automated measurements are increasing in use (Bongio et al. 2021; Kinar and Pomeroy 2015; Kopp et al. 2019), in-situ measurements remain the mainstay of data collection research and operations (Kinar and Pomeroy 2015; Pirazzini et al. 2018). Manual in-situ snow depth measurements are typically made using snow stakes, rulers, or narrow diameter snow probes (Kinar and Pomeroy 2015; Pirazzini et al. 2018). Snow tube samplers, which have been in use since the 1930s, also measure snow depth. The magnaprobe, an automatic snow depth probe that records snow





depth and GPS measurements, has considerably increased the number of georeferenced snow depth observations that can be made in a single day and is used extensively for snow depth research campaigns (Sturm and Holmgren 2018; Walker et al. 2020). Measurement variability and errors are sometimes reduced by repeating the measurement, typically three times (Leppänen et al. 2016). Because snow depth is assumed to have greater spatial variability than snow density (Elder et al. 1998), a snow survey often makes numerous snow depth measurements per snow density measurement then combines to obtain snow water equivalent (SWE) (López-Moreno et al. 2013).

SWE measurement errors associated with snow tube samplers are relatively well understood and characterized. Known issues include biases as compared to snow pit measurements (Dixon and Boon, 2012; Farnes et al., 1983; Goodison, 1978; Sturm et al., 2010), accuracies around +/- 5% to 10% for an individual instrument, and differences among SWE from different snow tube models (e.g., the Meteorological Service of Canada, the Federal or Mt. Rose, the Adirondack, and the Snow-Hydro) that can exceed 10% (Farnes et al. 1983). Less is understood about the errors in snow depth measurements. Lopez-Moreno et al.'s (2020) comparison of nine snow core samplers found that snow depths were relatively consistent when taken over a paved surface. However, over uneven ground, the snow depth differences among samplers was much greater and replicate snow depth measurements had larger variability as compared to the snow density. The magnaprobe, which measures snow depth with a precision of less than 0.1 mm, has the potential for low biases if its basket settles into soft surface snow, but those biases are typically less than 1 cm (Sturm and Holmgren 2018). When the rod penetrates the substrate (over-probing), the error depends on the ground surface and the operation. Solid or frozen ground surfaces have negligible over-probing, but unfrozen natural surfaces may have considerable penetration (Derry et al. 2009) with biases on the order of 5 to 10 cm (Berezovskaya and Kane 2007; Sturm and Holmgren 2018). These errors can have profound effects on SWE estimates in shallow snow environments and represent a challenge for error accounting in hydrological modelling.

The goal of this brief study is to determine 1) if the magnitude of the snow depth measurements using a magnaprobe and a Federal tube are significantly different in an ephemeral snow environment with shallow snow and 2) if the differences vary by land cover type. We hypothesize that the snow depth measurements from the magnaprobe will be deeper than those from the snow tube. This hypothesis is based on the understood errors and biases associated with each the magnaprobe and the Federal tube, including the smaller surface area of the probe which allows for greater penetration through snowpacks and leaf litter. Three snow depth sampling campaigns were conducted from December 2020 to March 2021 over field and forest plots at Thompson Farm in Durham, New Hampshire, USA.

## 2 Site, Methods, and Data

### 2.1 Study Site

This study was conducted at the University of New Hampshire's Thompson Farm Research Observatory in southeast New Hampshire, United States (N 43.11°, W 70.95°, 35 m above sea level, ASL). The site has mixed hardwood forest and open field land covers (Perron et al. 2004) that are characteristic of the region (**Fig. 1**). The agricultural fields are managed pasture grass with unmown grass in local areas. The deciduous, mixed, and



coniferous forest is composed primarily of white pine (*Pinus strobus*), northern red oak (*Quercus rubra*), red maple (*Acer rubrum*), shagbark hickory (*Carya ovata*), and white oak (*Quercus alba*) (Perron et al. 2004). The forest soils are classified as Hollis/Charlton very stony fine sandy loam and well-drained; field soils are characterized as Scantic silt-loam and poorly drained.

In-situ sampling was conducted at 39 sites located along three parallel transects (**Fig. 1**). The approximately 145 m long transects were laid out from east to west. The transects were separated by approximately 10 m, north to south. From east to west, each transect started in the open field area, then transitioned to the coniferous, then mixed, and finally, deciduous forested areas. Each of the three transects had 13 sampling sites; four sites were in the open field area, three in the coniferous forest, three in the mixed forest, and three in the deciduous forest, which were each marked with a stake. The stake locations were geolocated using a Trimble© Geo7X GNSS Positioning Unit and Zephyr™ antenna with an estimated horizontal uncertainty of 2.51 cm (standard deviation 0.95 cm) and 4.17 cm (standard deviation 4.60 cm) for the field and forest, respectively, after differential correction. Three soil frost tubes were located in the field approximately 25 m south of the field transect and another three in the forest about 100 m southwest of the study area.

**2.2 In-Situ Sampling Methods**

Snow depth was measured using a magnaprobe and a Federal snow sampler, also known as a snow tube. The Federal snow tube with its long operational history (Clyde 1932) served as a historical reference against the magnaprobe. A magnaprobe consists of an avalanche probe-like rod of about 1.5 m in length that contains a magnetostrictive device and a sliding magnetic disk-shaped basket with a 25 cm diameter. The rod has a 1.27 cm diameter with an affixed tip that tapers to a point to help penetrate ice layers. The magnaprobe was operated by inserting the pole into a snowpack until the tip of the pole reached the ground surface, allowing the basket to slide down to float on top of the snow. A handheld portable keypad connected to a datalogger recorded the snow depth between the tip of the pole and the bottom of the basket.

A Federal snow sampler is an aluminium tube, about 76 cm in length with a 4.13 cm inner diameter, that is used to measure snow depth and SWE (Clyde 1932). To measure snow depth, the snow tube was inserted vertically into the snowpack until it reached the ground, and a depth was read at eye level. Snow depth was recorded to the nearest 0.5 cm. To measure snow density, the snow tube was then lifted out of the snowpack, using a spatula as needed to ensure that snow did not fall out of the tube. The snow and snow tube were weighed using a digital hanging scale (CCi HS-6 Electronic Scale, 2-gram resolution).

Sampling campaigns were conducted on 18 December 2020, 4 February 2021, and 24 February 2021. A total of 936 paired magnaprobe and Federal snow tube snow depth observations were collected during the three sampling campaigns. At each of the 39 sampling locations, nine measurements were made in a 1x1 m area. At each location, a 1x1 m square polyvinyl chloride (PVC) grid was placed on the snow surface with one vertex located coincident with a stake. The orientation of two adjacent sides of the grid was recorded using a compass. Nine magnaprobe depth measurements were made at an approximately even spacing within the grid. Immediately after the magnaprobe measurements, snow tube snow depth measurements were made at the same nine locations by



115 positioning the snow tube directly over each magnaprobe sampling location. At a 10[th] location within each 1x1 m

116 grid, the snow tube was used to make a snow density measurement. For the 24 February 2021 campaign, after the

117 magnaprobe measurements were completed for the two northern transects, the instrument was transferred to a

118 new operator who made measurements on the southernmost transect (Transect 1). Transect 1 data for that date

119 were removed from the analysis because the QA/QC process identified notable errors for observations from that

120 transect.

121

122 Moultrie Wingscapes Birdcam Pro Field Cameras were used to capture images of the snowpack every 15 minutes

123 relative to a 1.5 meter marked PVC pole following the method used in NASA's 2020 SnowEx field camera

124 campaign in Grand Mesa, CO (personal communication, 16 November 2020). Three cameras were used; one was

125 in the open field, one was in the coniferous forest, and one was in the deciduous forest (**Fig. 1**). Snow depth was

126 derived by manual inspection of the photos and recorded to the nearest cm.

127 **2.3 Ancillary Soils and Vegetation Cover Data**

128 Daily soil frost depth data were collected at field and forest locations at the Thompson Farm Research Observatory

129 using Cold Regions Research and Engineering Laboratory style frost tubes (Gandahl 1957). The frost tubes have

130 flexible, polyethylene inner tubing filled with methylene blue dye whose color change is easy to differentiate

131 when extruded from ice. The outer tubing consists of PVC pipe installed between 0.4 to 0.5 m below the soil

132 surface. The field and forest sites each had three soil frost tubes.

133

134 Leaf litter depth was measured on 2 April 2021 after the spring snowmelt. The leaf litter depth was measured at

135 each snow depth sample location. Sampling was conducted using a PVC collar or round ring that is 8 cm in depth

136 and 10 cm in diameter (Kaspari and Yanoviak 2008). The collar was placed in the leaf litter and was pushed down

137 until it was through the leaf litter layer. If sticks or larger stones were in the way, they were either carefully

138 removed or the collar was moved slightly to an adjacent location. Measurements were taken using a wooden ruler

139 at four cardinal points in the collar. The four measurements were recorded and their average to the nearest cm

140 was used as the final litter depth. The range of leaf litter depths measured in the forest using the collar was typically

141 3 to 7 cm with an average leaf litter depth of 3.9 cm.

142 **3 Results**

143 The three sampling campaigns, 18 December 2020, 4 February 2021, and 24 February 2021, all had shallow

144 snowpacks. The snowpacks had similar depths, between 10 and 15 cm, on the three sampling dates with modestly

145 deeper snow in the field than the forest. The deepest snow was on 4 February 2021 with 15 cm in the field and

146 9.3 cm in the forest. Between the 18 December and the 4 February sampling campaigns, there was a melt event

147 in which the entire 10 cm snowpack on 18 December ablated. The next significant snowfall event (15 cm) occurred

148 on 1 February 2021. The snowpack experienced little additional accumulation or ablation between 4 February and

149 24 February. The 4 February (0.15 g/cm$^3$) and 24 February (0.20– 0.24 g/cm$^3$) snowpack density values were

150 higher than those in December (~ 0.10 g/cm$^3$). There were shallow soil frost depths (< 4 cm) during the early



winter 18 December campaign in the forest and the field. Deeper soil frost depths of 15.1 cm in the field and 5.9
cm in the forest occurred on 4 February 2021, with similar soil frost conditions on 24 February 2021.

**3.1 Magnaprobe vs. Snow Tube**

The full experiment yielded individual 936 pairs of snow depth measurements from the snow tube and the
magnaprobe (**Fig. 2a**). For the comparison between measurement techniques, the orthogonal Deming regression
method was applied to consider measurement errors in both variables. Overall, there was moderate agreement (R
= 0.74) between the two datasets for all three sampling campaigns (**Table S1**). The snow depths measured by the
magnaprobe (14.9 cm average snow depth) were deeper than the snow tube (13.2 cm average snow depth) with
an overall bias of 1.7 cm. The magnaprobe snow depth was at least 0.5 cm deeper than the snow tube in 74% of
the 936 measurement pairs. Only 6.3% of the pairs had snow tube snow depths exceeding magnaprobe snow
depths by 0.5 cm or more. Conversely, 7.4% of the pairs' magnaprobe snow depths were over 5.0 cm deeper than
the snow tube. In eight pairs of measurements, when the magnaprobe measured snow depth greater than 15 cm,
the magnaprobe snow depths were more than double the snow tube snow depth.
The majority of the nine sampling locations in each grid had magnaprobe snow depth values that were deeper than
those measured using the snow tube. For all the grids, an average of 8.7, 7.7, and 7.0 out of the nine sampling
locations had deeper magnaprobe snow depths on 18 December 2020, 4 and 24 February 2021, respectively. As
hypothesized, the magnaprobe snow depth values were significantly greater than those measured using the snow
tube for 39 and 31 of the 39 sampling locations on 18 December 2020 and 4 February 2021, respectively, but only
11 out of the 26 sampling locations on 24 February 2021. The mean differences were 2.3, 1.4, and 1.6 cm, with
root mean square difference (RMSD) values of 3.0, 2.3, and 3.3 cm, on 18 December 2020, 4 and 24 February
2021, respectively, which is on the order of 15 to 25% of the overall depth observed during these campaigns.
Despite the biases, the average within cell snow depth variability was nearly identical for the magnaprobe and the
snow tube in the field (1.3 cm standard deviation for the magnaprobe). In the forest, the Magaprobe's 2.0 cm
within-cell standard deviation modestly exceeded the snow tube's 1.5 cm standard deviation. A slightly reduced
agreement was found on 24 February when there was a 1 to 4 cm thick ice layer at the bottom of the snowpack in
local depressions.
The overall agreement between the snow tube and magnaprobe was better when the nine measurements within a
single 1x1 m grid cell were averaged at each of the sampling locations (**Fig. 2b** and **Table S1**). There is a notable
improvement in grid cell statistics, and the correlation is stronger (overall R = 0.87), with slopes closer to one,
intercepts closer to zero, and the RMSD values reduced to 2.5 cm or less. Although averaging has no impact on
the overall bias, the range of differences among pairs narrowed. The difference between the magnaprobe and the
snow tube is typically constrained to less than 3 cm with a limited number of outliers. The magnaprobe snow
depth was at least 0.5 cm deeper than the snow tube in almost all grid cells (86.7%), but only three grid cells had
differences greater than 5 cm. Among the grid averaged magnaprobe snow depths, there were no instances in
which there was a doubling of snow depth when compared to the snow tube measurements.



**3.2 Magnaprobe vs. Snow Tube by Land type**
The magnaprobe and snow tube snow depths differ by land type, with the field having deeper snow and more
spatial variability than the forest land types (**Fig. 3**). Among the three forest types, the deepest snow was in the
deciduous-dominated forest, with mixed and coniferous forest having similar snow depths. The mean difference
between the magnaprobe and snow tube snow depths is a modest 1.3 cm in the field and 1.9 cm in the forest, with
differences of 1.9, 2.0, and 1.9 cm in the deciduous, mixed, and coniferous land types, respectively. However, the
differences between the magnaprobe and snow tube snow depths in the forest were higher on 18 December (2.5
cm), than on 4 February, and 24 February, 1.7, and 1.4 cm, respectively. Based on t-test results, the magnaprobe
measured significantly deeper snow depth compared to the snow tube in both the field and the forest regardless
of whether individual locations (p-value < 0.001) or grid cell average snow depths (p-value = 0.02) were used.
Based on Welch's adjusted ANOVA test, there are no significant differences in over-probing among forest land
types (p-value = 0.24). The RMSD values between the magnaprobe and snow tube snow depths are 3.0 cm (2.3
cm) and 2.5 cm (2.0 cm) for the forest and field sampling sites (grid average values), respectively. Thus, the
sampling method has a different impact in the field than the forest and the RMSD and bias values provide an
indicator of the different errors associated with in-situ measurements based on land type when used for model or
remote sensing validation.
**4 Discussion and Suggestions**
This study quantifies the differences between snow depth measurements made with a magnaprobe and with a
snow tube. The differences seem to be primarily associated with greater over-probing by the magnaprobe into
vegetation/organic layers and thawed soils. The result was that the magnaprobe snow depth measurements were
higher than snow tube measurements, with a greater difference in the forest than in the field. An average of 5 cm
bias occurred in the tundra matte during the Cold Land Processes Experiment (CLPX) Alaska campaign (Sturm
and Holmgren 2018). Also in the open tundra environment found a 7.6 cm average over-probe penetration for
approximately 40 cm deep snow (Canada 2018). Berezovskaya and Kane (2007) also noted over-probing of 5 to
9 cm with a magnaprobe as compared to a snow tube found a bias in northern Alaska for snow depths between 29
and 48 cm. In this study, the over-probing, 1.3 cm in the field and a 1.9 cm in the forest, was less than previous
studies probably due to the lower range of snow depth and different surface conditions as compared to previous
studies.

We also agree with Lopez-Moreno et al.'s (2020) finding that it is important to understand the snowpack and land
conditions for which an individual sampler was designed to select the most appropriate sampler. Understanding
leaf litter or vegetation depths and underlying soils may potentially reduce and help to account for the over-probing
errors of magnaprobe snow depth measurements. Sturm and Holmgren (2018) suggested that operators need to
learn to push a magnaprobe through snow, yet not penetrate it too deeply into underlying vegetation/organic layers
by developing a sense for the base of the snowpack. This recommendation may be difficult to implement (e.g.,
over soft vegetation) where the probe easily penetrates the vegetation and problematic if multiple operators apply
a different amounts of force (Berezovskaya and Kane 2007). If operators over-probe into the base of the (frozen)
soils, one option is to consistently measure the depths in the same way (which would be snow depth *plus*



vegetation) and then subtract typical vegetation depths in the study area from the depths. When leaf litter is
evident, penetration into the organic layer should also be considered. In this study, we found that the 2.0 cm snow
depth differences were approximately half of the end of winter forest leaf litter depth (3.9 cm).
As observed in this study, leaf litter and soil frost may differentially impact in-situ snow depth sampling methods.
The earliest sampling campaign had limited soil frost and likely reduced litter compaction. Distinct contributions
of forest leaf litter depth to magnaprobe and snow tube snow depths may occur because the narrow magnaprobe
fully penetrates the leaf litter and the larger diameter snow tube only partially penetrates the litter, or the
magnaprobe may only partially penetrate the leaf litter but the snow tube does not break through the leaf litter.
Partial penetration of the magnaprobe into the leaf litter layer (i.e., over-probing) may vary by the freeze-thaw
state of the duff layer and/or mineral soil layers beneath the leaf litter layer. The horizontally aligned, matted leaf
litter could also limit snow tube penetration. High spatial variability of leaf litter depth could also be a factor,
though this was not quantified here. Thus, the increased differences among in-situ methods in forested areas
observed in this study point to the particular importance of in-situ validation in forested areas and, more generally,
sampling with multiple methods in an area with a nonuniform underlying substrate.
In summary, there are three major suggestions from this work below.
1) With an ephemeral snowpack in forested environment, snow depth measurements using a Federal
snow tube likely avoid over-probing that can frequently occur when a magnaprobe is used.
2) The use of the average of multiple point samples within a grid is recommended instead of single
measurements, because the average of multiple point samples can reduce the point-to-point variability
and spatial representativeness errors.
3) Measurements of vegetation, leaf litter, and soil frost can help to account for the errors of in-situ snow
depth observations, particularly when using a magnaprobe.
**5 Conclusion**
Manual in-situ sampling snow depth measurements can be made quickly and easily, but making consistent,
representative, and unbiased measurements can be challenging when the surface is irregular, vegetation/organic
layers and unfrozen soils result in over-probing, and the leaf litter compacts during the winter. This study
quantified the differences between snow depth measurements made with a magnaprobe and a Federal snow tube
in a mixed-use temperate forest landscape with ephemeral snowpack. For all sampling campaigns and land cover
types, the magnaprobe snow depth measurements (mean 14.9 cm) were usually, but not always deeper than the
snow tube measurements (13.2 cm) and had a 1.7 cm or 12% average difference. Biases were significantly higher
in the forest (1.9 cm) than the field (1.3 cm). The difference between the two instruments was 50% higher in early
winter campaign than the later campaigns. The differences among measurement techniques in this present study
reflect the current study area, surface conditions for a single season, and the operation of the instruments by this
project team. Further studies to understand the errors from in-situ sampling using snow probe are warranted in
various snow environments with different vegetation and soil conditions to provide guidance on best practices for
using in-situ snow probe datasets under conditions when over-probing is likely.



**Acknowledgements**

This material is based upon work supported by the Broad Agency Announcement Program and the Cold Regions Research and Engineering Laboratory (ERDC-CRREL) under Contract No. W913E518C0005 and W913E521C0006. The authors are grateful to Lee Friess for providing a technical review of the draft manuscript, Mahsa Moradi Khaneghahi for supporting manuscript preparation, and Brigid Ferris for training the team on litter depth sampling. Christina Herrick for post-processing GPS data.

**Data Availability**

The in-situ snow observations are available in supporting information.

**Author Contributions**

HP, JJ, EB, AH, FS, MP, and EC designed the research.  HP, CW, JJ, AH, FS, MP, EB, and EC conducted field work to obtain lidar and/or in-situ snow observations. HP, CW, JJ, EB, AH, and MP and performed the analysis. HP, EC, and AH produced the figures. HP, JJ, EB, and EC wrote the initial draft. All authors contributed to manuscript review and editing.

**Competing Interests**

The authors declare that they have no conflict of interest.

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



**Figure 1:** The 4 February 2021 aerial optical image of Thompson Farm, Durham NH, USA showing both forest
and field region with snow sampling sites in the field, coniferous, mixed, and deciduous forested areas as well as
the locations of the CRREL soil frost tubes; and field cameras.

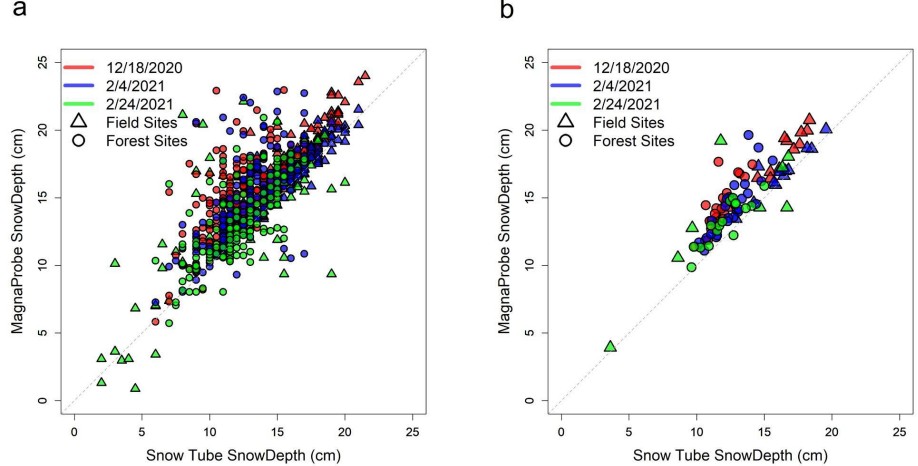


**Figure 2:** Comparison of snow depths measured by magnaprobe and snow tube for the three sampling campaigns
using (a) the sampling individual points (n = 936) and (b) using grid cell average values (n=104).

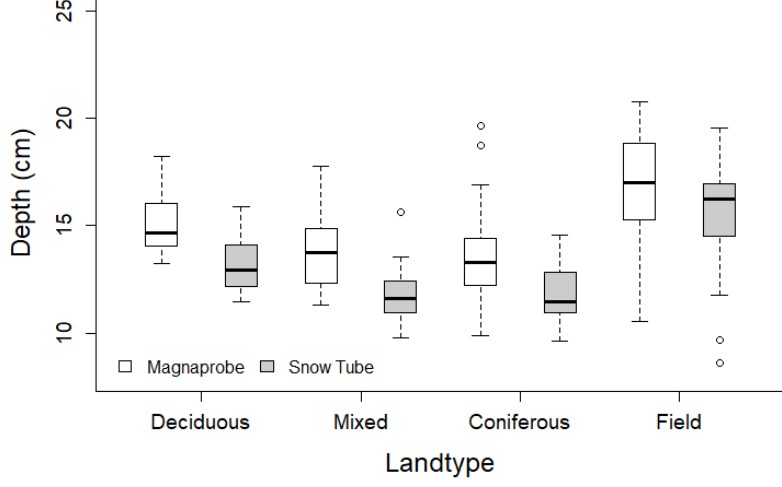

324 .

**Figure 3**: Boxplots of snow depths by land type measured by the magnaprobe and the snow tube for the three
sampling campaigns using the grid cell average values.