# Peer review of "Brief communication: Comparison of in-situ ephemeral snow depth measurements over a mixed-use temperate forest landscape"

_The Cryosphere, 2023_

## Referee Comment (RC1)

**Review comments** on tc-2023-36 manuscript, entitled," Brief communication: Comparison of in-situ ephemeral snow depth measurements over a mixed-use temperate forest landscape".

**General comments**:

The tc-2022-7 manuscript, entitled," Brief communication: Comparison of in-situ ephemeral snow depth measurements over a mixed-use temperate forest landscape" presents comparison snow depth measurements from two manual instruments (a magnaprobe snow depth probe and a federal snow tube) in a field and forest. A total of 936 paired observations are compared from sampling campaigns from December 2020 to February 2021.

The objectives of the study are to investigate the snow depth measurements using a magnaprobe and a federal tube are in an ephemeral snow environment with shallow snow and analyse differences whether related by land cover type.

As a general comment, the manuscript is designed and written well. It consists of extensive and important number measurements conducted and provided.

I would like to see and wonder following points,

- In line 125: "Snow depth was derived by manual inspection of the photos and recorded to the nearest cm." How are these snow depth measurements used? were they compared with magnaprobe and federal tube measurements?
- In line 134: "Leaf litter depth was measured on 2 April 2021 after the spring snowmelt. The leaf litter depth was measured at each snow depth sample location." There has been discussion on the effect of these litter depths with a different instrument. Since the measurements were done and I would expect to see some results or at least discussion on how these measurements can be used to calibrate the snow depth measurements with these two instruments (magnaprobe and deferral tube).

Please check references : I could not see "Lopez-Moreno et al.'s (2020)" in references

---

## Author Response (AR1)

**Response to A.N. Arsen Review**

**Review comments** on tc-2023-36 manuscript, entitled," Brief communication: Comparison of in-situ ephemeral snow depth measurements over a mixed-use temperate forest landscape".

**General comments**:

The tc-2022-7 manuscript, entitled," Brief communication: Comparison of in-situ ephemeral snow depth measurements over a mixed-use temperate forest landscape" presents comparison snow depth measurements from two manual instruments (a magnaprobe snow depth probe and a federal snow tube) in a field and forest. A total of 936 paired observations are compared from sampling campaigns from December 2020 to February 2021.

The objectives of the study are to investigate the snow depth measurements using a magnaprobe and a federal tube are in an ephemeral snow environment with shallow snow and analyse differences whether related by land cover type.

As a general comment, the manuscript is designed and written well. It consists of extensive and important number measurements conducted and provided.

A. Thank you for your thoughtful review and comments. Responses appear following each comment starting with "A."

I would like to see and wonder following points,

- In line 125: "Snow depth was derived by manual inspection of the photos and recorded to the nearest cm." How are these snow depth measurements used? were they compared with magnaprobe and federal tube measurements?

A. The field cameras snow depth values were used to observe the snow depth on the sampling dates and the snowpack evolution between dates. Because they only recorded a single location that was not coincident with the magnaprobe and federal tube measurements, we solely used these observations to qualitatively track the evolution of snow depth in the field and forest at consistent locations. The first paragraph of results was modified to read:

"The three sampling campaigns, 18 December 2020, 4 February 2021, and 24 February 2021, all had shallow snowpacks. The field camera observations indicate that the snowpacks had similar depths, between 10 and 15 cm, on the three sampling dates with modestly deeper snow in the field than the forest."

- In line 134: "Leaf litter depth was measured on 2 April 2021 after the spring snowmelt. The leaf litter depth was measured at each snow depth sample location." There has been discussion on the effect of these litter depths with a different instrument. Since the measurements were done and I would expect to see some results or at least discussion on how these measurements can be used to calibrate the snow depth measurements with these two instruments (magnaprobe and deferral tube).

A. Thank you for the opportunity to expand on this point. We also made measurements of leaf litter using the magnaprobe where the weight of the probe was the only force applied on the ground to minimize

penetration into the duff layer and underlying soil. Their average values were deeper than the collars. We did not make leaf litter depth measurements with the tubes.

We made minor modification to the methods, results, and discussion to add additional detail and to provide a more specific recommendation.

In the methods we added the magnaprobe information "Magnaprobe leaf litter penetration depth measurements, also made on 2 April 2021 in the forest, had an average value of 5.8 cm."

In Results section 3.2, we added a bit more information about the litter depth relative to the difference between the instruments. "While these differences are significant, the average litter depths exceeded the differences between the magnaprobe and snow tube snow depths in the forest, which were 2.5, 1.7, and 1.4 cm on 18 December, 4 February, and 24 February, respectively."

In the discussion, we recommended "When leaf litter is evident, penetration into the organic layer should be quantified by using independent leaf litter measurements, preferably using the snow depth sampling instrument, and use to bias correct snow depths."

Please check references: I could not see "Lopez-Moreno et al.'s (2020)" in references

   A.  Thank you. Added.

Response to Ryan Webb Review

Thank you for the thoughtful review and overall supportive feedback which has made this a more valuable contribution. Responses appear following each comment (A.)

This brief communication compares depth measurements of a magnaprobe and federal sampler in ephemeral snow conditions. The study is aimed at identifying the over-probing depth of the magnaprobe for the conditions studied. This is certainly an important topic due to the use of depth probes by so many snow scientists. Furthermore, overprobing has really only been done in the arctic as far as I know.

Overall, I really like the study and think a brief communication is a very appropriate format. The authors at least touch on many of the thoughts I had while reading the manuscript and mention suggestions for a path forward. Below are some specific comments listed by line number:

42: yes, depth is more variable, but when depth is well constrained (which this study will help to improve upon) density modeling becomes the source of uncertainty.

Raleigh, M. S., and Small, E. E. (2017), Snowpack density modeling is the primary source of uncertainty when mapping basin-wide SWE with lidar, *Geophys. Res. Lett.*, 44, 3700– 3709, doi:10.1002/2016GL071999.

 A.  Thank you for the point and the reference with implications for this work and lidar snow depth mapping, The introduction was modified as follows "If depth can be well constrained, then density becomes the source of uncertainty (Raleigh and Small 2017)."

104-105: So did you dig out the fed. sampler to use the spatula or did you pull a soil plug? Digging it out would provide further evidence of the Fed. sampler being a good reference depth.

 A.  We used the spatula only at the location that we measured density (the last observation in a grid). Because we were trying to measure depth, rather than density we did not want to penetrate the litter/soil excessively.

115: given the snow densities reported later, I am curious as to how much the magnaprobe disk compacted the snow surface. Wouldn't this decrease the depth measurement of the fed. sampler since taken at the same place after the magnaprobe? Did you account for this somehow?

A. Good point. We did not make compaction measurements but have a few photos. The image shown below from the field is probably the deepest compaction that we saw (December with 10% density). For reference, the PVC is 0.75" (O.D. 1.050" or 2.7 cm), so at most 20-25% or on the order of 0.6 cm. This agrees well with the Sturm and Holmgren (2018) finding that the magnaprobe has the potential for low biases if its basket settles into soft surface snow, but those biases are typically less than 1 cm. You can also make out in the image the unevenness of the snow surface. In most case where there was compaction, it was only part of the disk with less compaction at the center where the magnaprobe and the tube were coincident. Our interpretation is in most cases where there was no compaction, it was not an issue and where there was some compaction, the compaction was quite modest and the change in the surface elevation was similar to the compaction.

[Figure]

125: I know the manuscript format is limited in figures, but perhaps a camera image could be another panel in figure 1. It would be helpful to see what the landscape looks like on the ground. An image of the soil would be nice too, but that might be asking for too much when limited to 3 figures. Maybe supplementary info.?

A. We added landscape pictures taken by the field cameras from the field and the forest sites to Figure 1.

[Figure]

**Figure 1:** (a) The 4 February 2021 aerial optical image of Thompson Farm, Durham NH, USA showing both forest and field regions with snow sampling sites in the field, coniferous, mixed, and deciduous forested areas as well as the locations of the CRREL soil frost tubes; and field cameras. (b) Field camera images in the field, coniferous, and deciduous forested areas taken on 4 February 2021 by the field cameras.

162-163: Could the federal sampler also potentially be stopped by a branch protruding from a buried downed tree where the magnaprobe goes past the branch? less than 1% of measurements makes sense as this would be quite the rare occurence.

A. Absolutely. I think that gets to the heart of recommendation 2 - Use the average of multiple point samples within a grid. A comment was added to the discussion "Though in some cases the large differences could instead be due to the larger diameter snow tube hitting a branch from a down tree or debris that the magnaprobe bypassed."

168: What was the statistical test used for this significance?

A. We used a t-test after testing for normality. The results were modified to read "As hypothesized, t-test results showed that the magnaprobe snow depth values

were significantly greater than those measured using the snow tube for 39 and 31 of the 39 sampling locations on 18 December 2020 and 4 February 2021, respectively, but only 11 out of the 26 sampling locations on 24 February 2021."

176: Were you able to confirm if the magnaprobe or fed sampler were going through the ice layer or if one was not and the other was?

   A. We were not able to confirm.

198-199: No differences among land types, but what about based on user? Not sure if you have enough data to test that or not, but a thought that I had.

   A. We did not test difference between users and made sure that the same individual operated the instrument for each campaign. Our impression matches yours – switching individual mid-campaign or trying to compare across sites would introduce a difference. One consideration would be to have both individual sample at a single location to quantify any differences.

258: I suggest avoiding the % difference value in the conclusion simply to avoid someone mis-understanding and applying a 12% correction to a deeper snowpack when it should only be a ~1.7 cm correction.
   A. You make a good point. We are trying make the point that these difference matter in a shallow snowpack, but probably do not matter as much in a deeper snowpack. We rewrote the section by removing the 12% from the original sentence and adding another sentence that is less likely to be misunderstood. "For all sampling campaigns and land cover types, the magnaprobe snow depth measurements (mean 14.9 cm) were usually, but not always deeper than the snow tube measurements (13.2 cm) and had a 1.7 cm average difference. For these shallow snowpacks, this amounts to a 12% difference, but in a deeper snow pack the relative impact of this difference would be much smaller."